# Sensitive Detection of Industrial Pollutants Using Modified Electrochemical Platforms

**DOI:** 10.3390/nano12101779

**Published:** 2022-05-23

**Authors:** Alessio Di Tinno, Rocco Cancelliere, Pietro Mantegazza, Antonino Cataldo, Alesia Paddubskaya, Luigi Ferrigno, Polina Kuzhir, Sergey Maksimenko, Mikhail Shuba, Antonio Maffucci, Stefano Bellucci, Laura Micheli

**Affiliations:** 1Department of Chemical Science and Technologies, University of Rome “Tor Vergata”, 00133 Rome, Italy; alessio.ditinno@gmail.com (A.D.T.); roccocancelliere@gmail.com (R.C.); mantegazza.pietro@gmail.com (P.M.); 2DISPREV Laboratory, Casaccia Research Center, ENEA, 00185 Rome, Italy; antonino.cataldo@enea.it; 3National Institute of Nuclear Physics, Frascati National Laboratories, 00044 Frascati, Italy; stefano.bellucci@lnf.infn.it; 4Institute for Nuclear Problems, Belarusian State University, 220007 Minsk, Belarus; paddubskaya@gmail.com (A.P.); sergey.maksimenko@gmail.com (S.M.); mikhail.shuba@gmail.com (M.S.); 5Department of Electrical and Information Engineering, University of Cassino and Southern Lazio, 03043 Cassino, Italy; ferrigno@unicas.it (L.F.); maffucci@unicas.it (A.M.); 6Department of Physics and Mathematics, Institute of Photonics, University of Eastern Finland, 80200 Joensuu, Finland; polina.kuzhir@uef.fi

**Keywords:** carbon-based nanomaterials, graphene nanoplatelets, organic pollutants, quinones, screen-printed electrodes, voltammetry

## Abstract

Water pollution is nowadays a global problem and the effective detection of pollutants is of fundamental importance. Herein, a facile, efficient, robust, and rapid (response time < 2 min) method for the determination of important quinone-based industrial pollutants such as hydroquinone and benzoquinone is reported. The recognition method is based on the use of screen-printed electrodes as sensing platforms, enhanced with carbon-based nanomaterials. The enhancement is achieved by modifying the working electrode of such platforms through highly sensitive membranes made of Single- or Multi-Walled Carbon Nanotubes (SWNTs and MWNTs) or by graphene nanoplatelets. The modified sensing platforms are first carefully morphologically and electrochemically characterized, whereupon they are tested in the detection of different pollutants (i.e., hydroquinone and benzoquinone) in water solution, by using both cyclic and square-wave voltammetry. In particular, the sensors based on film-deposited nanomaterials show good sensitivity with a limit of detection in the nanomolar range (0.04 and 0.07 μM for SWNT- and MWNT-modified SPEs, respectively) and a linear working range of 10 to 1000 ppb under optimal conditions. The results highlight the improved performance of these novel sensing platforms and the large-scale applicability of this method for other analytes (i.e., toxins, pollutants).

## 1. Introduction

Over the last half-century, the disproportionate industrialization has caused a significant increase in the release of potentially toxic compounds, thus contributing considerably to the “great challenge” of our day: environmental pollution. Industrial waste may be solid, semi-solid, or liquid in form and may pollute the nearby soil or adjacent water bodies, contaminating groundwater, lakes, streams, rivers, or coastal waters [1]. Nowadays, most countries have enacted legislation to deal with the problem of industrial waste, but strictness and compliance regimes vary, not always being effective [2]. Moreover, recent research has shown that environmental pollutants are directly connected to the increase in human diseases, particularly those involved with the immune system [3,4]. In this regard, the contribution of benzene and its metabolites, such as quinones, to this issue is well recognized, making them a public health problem [5,6]. Among the benzene metabolites, hydroquinone (HQ) and benzoquinone (BQ) are undoubtedly two of the most important, due to their widespread application in human and industrial activities (i.e., as an inhibitor, electrochemical mediator, antioxidant, reducing agent, and intermediate in the synthesis of dyes, motor fuels, and oils, in photographic processing, and in cosmetics and medical preparations) [7,8,9,10,11,12,13,14,15,16,17,18,19,20,21,22,23]. HQ and BQ, also called 1,4-Benzenediol and Cyclohexa-2,5-diene-1,4-dione, respectively, are aromatic compounds comprising a benzene core carrying two hydroxyl/chetonic substituents at a para position to each other. They occur in the environment as a result of anthropogenic processes (i.e., industrial productions), as well as in natural products from plants and animals [10] (i.e., as arbutin, a glucose conjugate, or in wheat, pears, *Blaps lethifera*, *Pyrus*, coffee, onion, tea, and red wine) [10,24,25,26,27,28,29]. Long exposure to HQ and BQ can cause different types of health problems, from irritation (skin, eyes, nose, and throat) to mutagenic effects (in animals) and even acute poisoning, which can cause dizziness, headache, loss of consciousness, difficulty breathing, and a fast or weak pulse [4,30,31,32,33,34,35]. This is exactly why HQ and BQ were classified as harmful chemicals on the Approved Supply List and are labeled with risk phrases.

A lot of research has been carried out and several methods for the determination of these quinones have been developed [16]. In particular, numerous scientific publications based on chromatographic and spectrophotometric analysis (i.e., liquid chromatography–UV detection, HPLC–MS) are reported in the literature [36,37,38,39]. For example, Gimeno and his coworkers developed an HPLC–UV-based method for identifying and screening HQ (LOD in the μM range) in cosmetic products [38]. However, there are other works, based on liquid chromatography, which aimed to detect HQ and BQ in pharmaceuticals, gel and cream preparations, water, and cosmetics [40,41,42,43]. Spectrophotometric methods have also been used [44,45,46,47,48,49]. Although these methods to measure HQ and BQ levels have been demonstrated to be effective, they still require a minimum of in-presence sample processing and high employment of reagents and time [50,51]. In this context, electrochemical sensors represent a valid alternative to the traditional staple techniques due to their instrumental simplicity, cost-effectiveness, and portability [50,52]. Indeed, several works have been reported in the literature where electrochemical methods are used for the determination of quinones. For example, Cotchim et al. (2020) developed an electrochemical sensor based on a carboxylic acid-functionalized graphene (Gr-COOH)-modified glassy carbon electrode for the determination of HQ in pharmaceutical products (LOD of 0.1 μM) [53]. Another interesting work is that of Sawczuk and her coworkers, who developed an electrochemical nanosensor based on iron oxide nanoparticles and Multiwalled Carbon Nanotubes (MWNTs) for the simultaneous determination of benzoquinone and catechol in groundwater with a sensitivity in the micromolar range [54]. There are more, as well resumed by Guin et al. (2010) in their work [55]. Nevertheless, they all share one trait: the use of complicated setups with more than one modification step: for example, carbon nanomaterials conjugation with metal nanoparticles.

Herein, a facile and one-step-prepared sensor based on Carbon Nano-Materials (CNMs)-modified screen-printed electrodes (SPEs) is reported. It is well-known that the conventional SPEs (i.e., graphite, carbon) have serious issues due to their sluggish surface kinetics, which severely affects the sensitivity (i.e., broad peak, high potential needed, no peak at a lower concentration) and selectivity of the platforms [56,57]. Nowadays, CNMs represent the most valid solution to overcome this problem, owing to the wide range of applications and their physical and chemical properties [58,59,60]. Among the several advantages provided by the modification of SPEs with CNMs, the improved surface kinetics, the enhanced electroactive surface area, and the amended adsorption and functionalization capability are the most important [56,61,62]. Herein, we decided to use CNMs for the development of model sensors for the rapid, sensitive, and highly reproducible environmental monitoring of HQ and BQ. In particular, we report on the post-printing modification of the working electrode (WE) of our serigraphic platform by using three of the most promising types of graphene-based materials, specifically single- and multi-walled carbon nanotubes (SWNTs and MWNTs) and graphene nanoplatelets (GNPs). At first, we investigated electrochemically the electro-catalytic effectiveness (attributable to the π-conjugated honeycomb structure) of carbon nanotubes (CNTs) and GNPs, once used in the modification of SPEs. In particular, CNTs films were used to create and thus modify the working electrodes (WE) of SPEs, whereas the GNP-modified electrodes were prepared using bare SPEs drop-casted with a dispersion of GNPs. After that, the analytical performances of CNM-based sensors for the detection of the chosen quinones in buffer and real sample matrix (water) were tested. The results indicated that the presence of CNMs, particularly SWNTs, led to a significant increase in the intensity of the recorded Faradic current (resulting from the growth surface area of the platform) in comparison to bare SPEs. The repeatability values (RSD%) fluctuated from 11% of the bare SPEs to 6% of the SWNT-SPEs. In this case, therefore, the modified platforms were also configured as a suitable tool for the quantification of organic analytes present in low concentration in an aqueous solution (below 0.5 mM of hydroquinone).

## 2. Materials and Methods

### 2.1. Materials

All chemicals from commercial sources were of analytical grade. Ethanol, potassium ferri/ferrocyanide, p-benzoquinone, and hydroquinone were purchased from Sigma-Aldrich (Steinheim, Germany). MWNTs and SWNTs were purchased from Heji Inc. (Hong Kong) and Nanointegris Technology Inc. (Boisbriand, QC, Canada), respectively. GNPs were in-house-produced by microcleavage exfoliation of the expanded graphite (provided by Asbury^®^, Wilmore, KY, USA) and reported in detail in [59,62]. The buffer solution used was 0.05 M phosphate-buffered saline (PBS), 0.1 M KCl, pH = 7.4.

### 2.2. Fabrication of GNPs and CNTs and Integration into SPEs

The microwave exfoliation technique, a low-cost and industrially scalable procedure [59,62], was used to prepare the GNPs employed in the modification of our platforms. In particular, the procedure started from commercial expandable graphite (EG), intercalated with sulfates and nitrates (inserted between the adjacent layers of sp^2^-hybridized carbon). The graphite was then heated in a microwave oven (800 W power, 10 s). The thermal shock vaporized the molecules present inside the EG, expanding to form worm-like graphene. These structures were then placed in isopropyl alcohol and sonicated for about 10 min, obtaining a solution with the final GNPs of typical lateral sizes in the range of 2–10 μm, as in the example given in Figure 1a, where a Scanning Electron Microscope (SEM) image of a GNP is provided.

Structural characterization was carried out using Raman spectroscopy, see Figure 1b. Specifically, the analysis of the 2D bands led to the determination of the number of layers of these GNPs, *N_G_*, through the following relation [62]:(1)$$N_{G}=10^{0.84M+0.45M^{2}}\;,\;\;M=\frac{I_{G^{\prime }ene}\left(w=w_{p,G^{\prime }ite}\right)/I_{G^{\prime }ene}\left(w=w_{s,G^{\prime }ite}\right)}{I_{G^{\prime }ite}\left(w=w_{p,G^{\prime }ite}\right)/I_{G^{\prime }ite}\left(w=w_{s,G^{\prime }ite}\right)},$$
where $I_{G^{\prime }ene}$ and $I_{G^{\prime }ite}$ are the Raman intensities of graphene and graphite, respectively. The estimated number of layers was $N_{G}$ = 5 ± 1.5. From the analysis of the peaks, D and D’ good-quality and low-density defects on GNPs were observed, with the ID/ID’ ratio being equal to 3.9 ± 0.1. Finally, the value of ID/IG of about 0.05 revealed a mean distance between defects over 35 nm, thus indicating a low density of defects. Finally, an Pinfrared (IR) analysis confirmed that the proposed fabrication procedure did not introduce significant oxidation, thus preserving the conductive properties (see Appendix A).

The CNT films were fabricated from commercial MWNTs and SWNTs via a vacuum filtration technique. Briefly, 0.1 mg of the CNT material was suspended in 10 mL of an aqueous solution containing 1 wt% of sodium dodecyl sulfate (SDS) by ultrasonication treatment (44 kHz) for 1 h. Then, the suspension was centrifuged at 8000× *g* for 20 min. After centrifugation, the supernatant was filtered with a cellulose-ester membrane (Millipore, 0.22 µm pore size) by using a vacuum cell. During the filtration process, the CNTs accumulated onto the membrane filter surface forming a homogeneous film. The surfactant was partly washed away with 200 mL of distilled water at a temperature of 60 °C. The hot water helped to remove surfactant from the tube surface. Then, the filter was dissolved in acetone, which was used also to wash the CNT films. The acetone was substituted by ethanol, and the CNT film was transferred to the working electrodes (WE) of SPEs. The GNP-modified SPEs were prepared as follows. Initially, screen-printed electrodes were pretreated amperometrically (1.7 V, 180 s) using a 0.05 M phosphate buffer +0.1MKCl, pH 7. After that, the electrodes were rinsed with distilled water (to remove salt residues) before modifying those using GNPs (a suspension of 1 mg/mL of powder in 1:2 ethanol:water). An ultrasonic transducer (200 W, 26 kHz and 30 min) was used to homogenize the suspension. The working electrodes (WE) were modified by drop-casting with 6 μL of CNM dispersions. Once dried at room temperature, the GNPs-SPEs were ready to use. A schematic representation of the CNT film and GNP-based sensors is reported in Figure 2.

From a structural characterization, the film thickness for both SWNTs and MWNTs was estimated as 200 and 1000 nm, respectively. A SEM characterization of the obtained films showed no visible surfactant on the surface of the MWCNTs, see Figure 3a. Visually, the density of SWCNT film was higher than that of MWNT film. This could explain why the MWNT film was better washed from the surfactant than the SWNT film. In addition, the Raman spectra of MWNTs and SWNTs were studied (see Figure 3b). The ratio of the intensity of the D-mode to that of the G-mode for the MWNTs was high (0.87), indicating the defective nature of their crystalline structure. This ratio was low (0.08) for SWNTs, indicating the low quantity of sidewall defects in them.

We stress that the amount of the modification material influences the electrochemical performances of the electrode (see [63,64]). Therefore, here, the concentration of the nanomaterials used to modify the electrode was optimized by looking for the saturation amount able to ensure the reproducibility of the electrochemical performance of devices.

### 2.3. Apparatus

The electrochemical measurements were carried out using in-house-produced SPEs, as detailed in our previous works [63]. Cyclic voltammetry (CV) and square-wave voltammetry (SWV) analysis were performed using a Palmesens4™ portable potentiostat system (Palmesens, Houten, The Netherlands) together with proprietary software PSTrace (Palmesens, Houten, The Netherlands). Dispersions were realized using an Hielscher UP200St-Ultrasonic Transducer. Morphological analyses were performed using a VEGA II scanning electron microscope (Tescan, Czech Republic). Raman spectra were performed using an Invia Raman microscope (Renishaw, UK) endowed with a 532 and 633 nm laser, a 100× objective, and an 1800 L/mm grating

### 2.4. Analytical Parameters Calculation

The limit of detection and quantification (LOD, LOQ), the heterogeneous rate constants (k^0^), and the percentage increase (I%) in faradaic current were estimated as reported in our previous works [63,65]. Moreover, the Randles–Sevcik equation (Equation (2)), exploited for the determination of the diffusion coefficients (*D*_0_) and the surface-active area (*A*), is set forth below [66]:(2)$$I_{p}=\left(0.4463\right)nFAC\sqrt{\frac{nFvD_{0}}{RT}}$$
where *D*_0_ is the diffusion coefficient (cm^2^/s) for the ferri/ferrocyanide couple, ν is the scan rate (vs^−1^), *n* is the number of electrons involved in the process, *F* is the Faraday constant (1/mol), *T* is the temperature (K), *R* is the universal gas constant (J/K mol), and *A* is the active electrodic surface.

## 3. Results and Discussion

### 3.1. Electrochemical Performances of CNM-Modified Platforms

To demonstrate the effect of carbon-nanomaterials on the performance of our homemade serigraphic platforms, an in-depth electrochemical characterization of unmodified (bare SPEs) and modified SPEs (MWNTs, SWNTs, GNP-SPEs) was carried out (Figure 4). Initially, the background current output was quantified using amperometry (1.7 V, 150 s) by analyzing eight different electrodes (*n* = 8) for each type of platform in a 100 mM KCl solution. The following current results were obtained: 29.4 ± 0.5, 40 ± 1, 35 ± 9 nA, and 132 ± 14 nA for SWNTs, MWNTs, GNPs, and bare SPEs, respectively. By using the standard deviation [50] as the index of signal/noise ratio, it was possible to ascertain that the modification with CNMs produced a more dependable electrochemical device with an important minimization of capacitive current. After that, six different platforms (*n* = 6) for each modified-SPEs were analyzed voltammetrically (CV) using the reversible couple (10 mM) [Fe(CN)_6_]^3-/4-^ as an electroactive reporter. By analyzing the voltammograms reported in Figure 4a, it can be readily observed that the modification of the WE with CNMs produced a dramatic improvement in the magnitude of the voltammetric peak height (faradaic current), as proven by the % increase in anodic and cathodic peak current (Ipa and Ipc). In particular, a 19, 16, and 17-fold increase in the anodic peak current was observed for SWNTs, MWNTs, and GNP-based SPEs, respectively (Figure 4a). In addition, once drop-casted, the nanofunctionalized platforms allowed the overcoming of a serious issue that severely affects the sensitivity and selectivity associated with conventional electrodes: the sluggish surface kinetics. As reported in Table 1, an incredible improvement of the surface kinetics of the bare electrodes was achieved by modifying them with CNMs, as demonstrated by the heterogeneous electron transfer rate constant (k^0^) calculated here. Moreover, the nanomodified platforms allowed the electroactive couples (Fe^2+^/Fe^3+^) to function as an ideal reversible system, as evidenced by the anodic and cathodic peak ratio (Ipa/Ipc = 1) and the peak-to-peak separation (ΔE). By contrast, a quasi-reversible systems-like behavior was observed for bare SPEs. The enhanced surface kinetic is not the only advantage that the use of nanomaterials brings in electrode modification. A major electroactive surface area (A) is usually observed after the drop-casting of the working electrode (WE). In particular, a 2, 3, and 4-fold increase in A was observed when unmodified-SPEs were cast using SWNTs, MWNTs, and GNPs, respectively. The improvement of all these electroanalytical parameters formally entails a decrease in the background current and therefore an increase in the sensitivity-correlated faradic current signal. Indeed, CNM-modified SPEs became fivefold more sensitive than the bare platform when tested voltammetrically (SWV) using different concentrations (from 0 to 10 mM) of [Fe(CN)_6_]^3−/4^. Furthermore, the diffusivity process occurring at the electrode interface was studied. This was performed by evaluating the effect of the scan rate on the redox peak currents (Figure 3b). The voltammograms reported in Figure 4b revealed that the faster the scan rates, the smaller the size of the diffusion layer and, accordingly, increased peak current values were observed. In particular, using the Randles–Sevcik equation (Equation (2)), in which the peak current is directly proportional to the square root of the scan rate (γ), a plot reporting Ipa and Ipc as a function of the γ ^0.5^ was obtained (Figure 4c). The deriving slopes (μA/(mV/s)^0.5^) of the linear correlation of Ipa and Ipc were calculated: 174.5 (R^2^ = 0.999) and −174.7 (R^2^ = 0.999), 138,8 (R^2^ = 0.998) and −138.2 (R^2^ = 0.999), 137.7 (R^2^ = 0.997) and −138.2 (R^2^ = 0.995), and 121.7 (R^2^ = 0.997) and −120.4 (R^2^ = 0.997), corresponding, respectively, to the SWNT, MWNT, GNP, and bare SPE-based sensor. Moreover, by further manipulating [63] Equation (2), the diffusion coefficient (D_0_), as the average of the anodic and cathodic diffusion coefficient (D_Ox_ and D_Red_), was calculated. Precisely, ferro-ferricyanide was used as an electroactive couple using CV as voltammetric tools with a scan rate of 30 mV/s. By comparing the results reported in Table 1 with those of Konopka and McDuffie reported in our previous work [29,32], a very similar diffusional process was assessed, thus demonstrating a planar diffusion-controlled process in the oxidation/reduction reactions of the [Fe(CN)_6_]^−4/−3^ for the different nano-engineered platforms.

To ascertain the effective sensitivity of the CNM-modified platforms, a preliminary study analyzing different concentrations of [Fe(CN)_6_]^4−/3−^ by using SWV as an analytical technique was carried out. The limit of detection (LOD), the sensitivity, and the reproducibility, calculated by analysing the response of a 10 mM electroactive probe using six electrodes fabricated by the uniform procedure, are reported below (Table 1).

According to this preliminary electrochemical characterization, SWNT-based platforms resulted in 3, 6, and 22-fold more sensitivities than MWNT, GNP, and bare SPE-based sensors, respectively. Nevertheless, all the nanomodified platforms shared a good reproducibility (RSD ~ 5%) and surface kinetics, as demonstrated from the k^0^ of the same order of magnitude, thus demonstrating the effectiveness of modification and the significant improvement of the electrodic conductivity once modified.

### 3.2. CNM-Based Sensor for the Determination of Hydroquinone and Benzoquinone in Buffer Solution

To ascertain the analytical performances (i.e., sensitivity, reproducibility) of our nanomodified platforms in the determination of HQ, two different analytical techniques, CV and SWV, were used. In Figure 5, the relative results obtained by analyzing (by CV) several HQ concentrations (from 0 to 10 mM) using SWNTs, MWNTs, GNPs, and bare SPEs were reported. From a visual inspection of Figure 4a, it was possible to confirm that SWNT-based platforms resulted in the most performing and reproducible (RSD% < 5%). In particular, the linear regression (y = mx + q) shown in Figure 5e was obtained by reporting the concentrations of HQ as a function of the relative recorded currents in terms of the “current vs. concentrations” slope (m) (650.9, R^2^ = 0.997; 566.4, R^2^ = 0.995; 544.6, R^2^ = 0.995; and 268.2, R^2^ = 0.994, respectively) for SWNTs, MWNTs, GNPs, and bare SPEs. Furthermore, by using the standard deviation (obtained from the current recorded by analyzing ten different blank samples) and the slope reported above for each platform, the following limits of detection (LOD) were calculated: 8.5, 11.6, 15.8, and 90.3 μM for SWNTs, MWNTs, GNPs, and bare SPEs, respectively. Considering these results as good but improvable, we repeated these experiments (six electrodes for each concentration) using a more performing electrochemical technique such as the SWV (see Figure 6). As for the CV results, also in this case, the SWNT-modified SPEs resulted in the most performing platform (Figure 6a) in terms of sensitivity. Moreover, by calculating the LOD for all the proposed sensing devices, the following results were obtained: 0.04, 0.07, 0.3, and 12.3 μM for SWNT, MWNT, GNP, and bare SPE-based devices, respectively. In addition, benzoquinone (BQ), as a second important pollutant, was studied, using both CV and SWV, and the relative results are reported in Table 2. For BQ, as for HQ, excellent results in terms of LOD and reproducibility were obtained, thus demonstrating the applicability of these nano-engineered platforms as rapid, sensitive, and accurate lab-on-chip devices for pollutant determinations. The response time was the same as the measurement time (<2 min).

### 3.3. CNM-Based Sensor for the Determination of Hydroquinone and Benzoquinone in Water

To ascertain the applicability of the developed CNM-based platforms in a real matrix, a careful voltammetric (CV and SWV) study at different concentrations (from 0.1 μM to 10 mM) of HQ and BQ in water samples was carried out (voltammograms reported in Appendix A). The results (see Table 3) are very encouraging for both the pollutants employed, proving the efficacy of this method as a rapid environmental control system. In particular, the linear regression (m) obtained by analyzing the HQ-spiked water sample with CV and SWV, respectively, for all the sensors at our disposal was: 31.6 (R^2^ = 0.994) and 42.8 (R^2^ = 0.998), 27.9 (R^2^ = 0.994) and 31.3 (R^2^ = 0.997), 19.2 (R^2^ = 0.994) and 26.7 (R^2^ = 0.997), and 13.4 (R^2^ = 0.992) and 19.8 (R^2^ = 0.993) for SWNT, MWNT, GNP, and bare SPE-based sensors. By contrast, the slopes obtained for the BQ-spiked water sample were: 17.3 (R^2^ = 0.996) and 170.4 (R^2^ = 0.998), 10.1 (R^2^ = 0.995) and 120.2 (R^2^ = 0.994), 10.2 (R^2^ = 0.994) and 64.5 (R^2^ = 0.993), and 5.4 (R^2^ = 0.998) and 10.2 (R^2^ = 0.999) for SWNT, MWNT, GNP, and bare SPE-based sensors, respectively. As noted in the preliminary characterization (Section 3.1), the SWNT-based platforms resulted in the most performing devices in terms of analytical robustness. However, the modification of the electrodic surface with all our carbonaceous nanomaterials led to a validation of the RSD% values below the 15% threshold value (the accepted standard for serigraphic platforms).

### 3.4. Stability and Recovery of CNM-Based Sensors in HQ Determination

The stability of the CNM-based biosensors was examined by repeating the measurements (SWV) of 300 µM HQ periodically over five weeks. The investigation was realized for all the different platforms stored in a humid chamber at 4 °C without using any preservatives. The results in terms of registered faradic current showed an almost constant response for up to one month. Precisely, an averaged intra-day (six SPEs for each platform) and inter-day (from day 1 to 35) repeatability of 8% and 15% were calculated for all our platforms, respectively. Moreover, a recovery study using SWNTs, the most powerful platform, and bare SPE-based sensors was carried out by analyzing a series of known HQ concentrations spiked in water (1, 10, 50, 100 µM). The results reported in Table 4 were obtained following the procedure adopted in our previous works, e.g. [63,65].

The percent recovery values for the spiked samples were 85.0 to 92.8% and 5.0 to 79.2% for SWNT-SPEs and bare SPEs, respectively. Therefore, a negligible matrix effect and a good efficiency of the developed SWNT-SPEs were verified in the analysis of HQ in water samples. At the same time, it can be extrapolated from these data how better CNM-SPEs are if compared to bare ones.

## 4. Conclusions

This work proposed cost-effective and easy-to-build and -use disposable screen-printed-based electrochemical tools for the quantification of industrial pollutants, such as hydroquinone (HQ) and p-benzoquinone (BQ). Electrochemical characterization through cyclic (CV) and square-wave voltammetry (SWV) showed the improved performance of such platforms when modified by carbon nanomaterials. Among all, SWNT-film-based SPEs resulted in the best-performing sensing platforms in terms of sensitivity, repeatability, and signal-to-noise ratio, both in the preliminary study and in the determination of HQ and BQ.

In particular, when used in these analyses (SWV), LODs of 0.04, 0.05, 2.2, and 1.7 µM and reproducibilities (RSD%) of 5, 8, 10, and 11% for HQ and BQ in the buffer solution and real-spiked sample (water), respectively, were obtained. In addition, a quick response time (<2 min) was shown. These results demonstrated the applicability of these nano-engineered platforms as rapid, sensitive, and accurate lab-on-chip devices for pollutants determinations. Indeed, an incredible improvement was obtained by comparing the analytical performances of our platforms to bare SPEs (LOD of 132.3 µM for HQ and 121.2 µM for BQ). Lastly, a recovery study, carried out by using SWNT-SPEs compared to bare electrodes, revealed high percent recovery values (95 to 92.8%) for the spiked samples, proving once again the powerfulness of these platforms with a negligible matrix effect.

## Figures and Tables

**Figure 1 nanomaterials-12-01779-f001:**
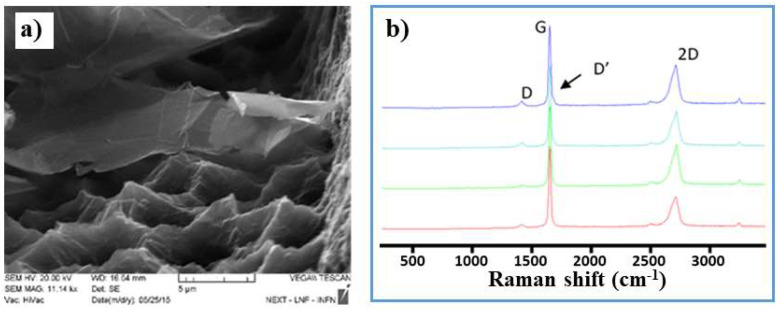
Structural characterization of the graphene nanoplatelets: (**a**) a SEM micrograph of GNP flakes; (**b**) Raman spectroscopy results.

**Figure 2 nanomaterials-12-01779-f002:**
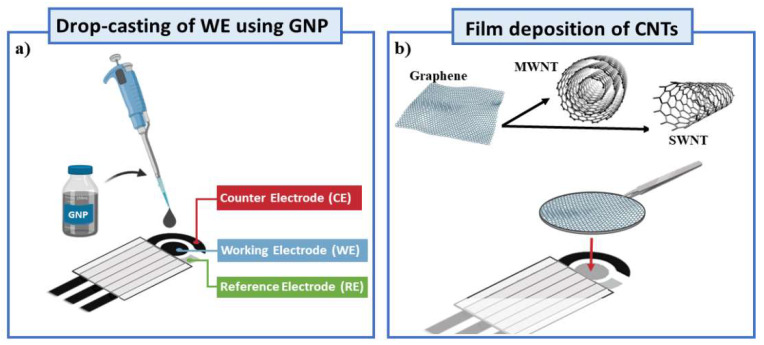
Schematic representation of nano-engineered-SPE. (**a**) Classical drop-casting of WE surface using a dispersion of GNPs, (**b**) innovative deposition of CNT-based films.

**Figure 3 nanomaterials-12-01779-f003:**
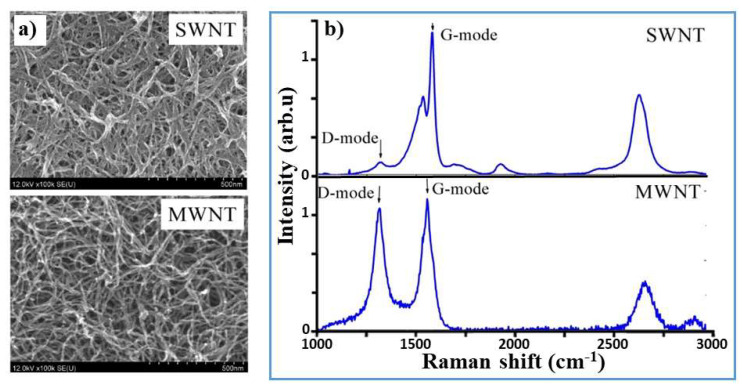
Structural characterization of the CNTs: (**a**) SEM micrographs of the SWNT and MWNT films; (**b**) Raman spectroscopy results.

**Figure 4 nanomaterials-12-01779-f004:**
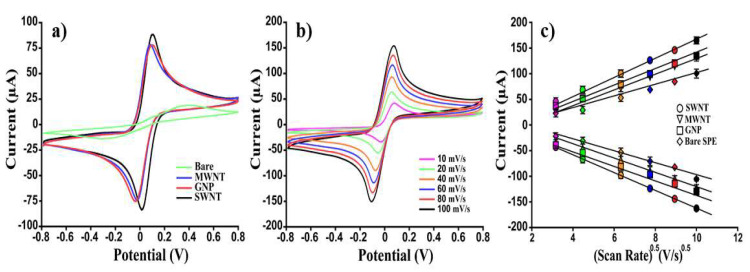
Electrochemical characterization of electrode interface after CNM modification. (**a**) Comparison of the cyclic voltammograms, (**b**) scan rate study (for MWNT-based SPEs), and (**c**) linear regression of SWNTs, MWNTs, GNPs, and bare SPEs obtained using 10 mM [Fe(CN)_6_]^4-/3-^ in 0.05 M PBS. Curves of one representative CNM-modified SPE sensor of at least 6 analyzed platforms are presented.

**Figure 5 nanomaterials-12-01779-f005:**
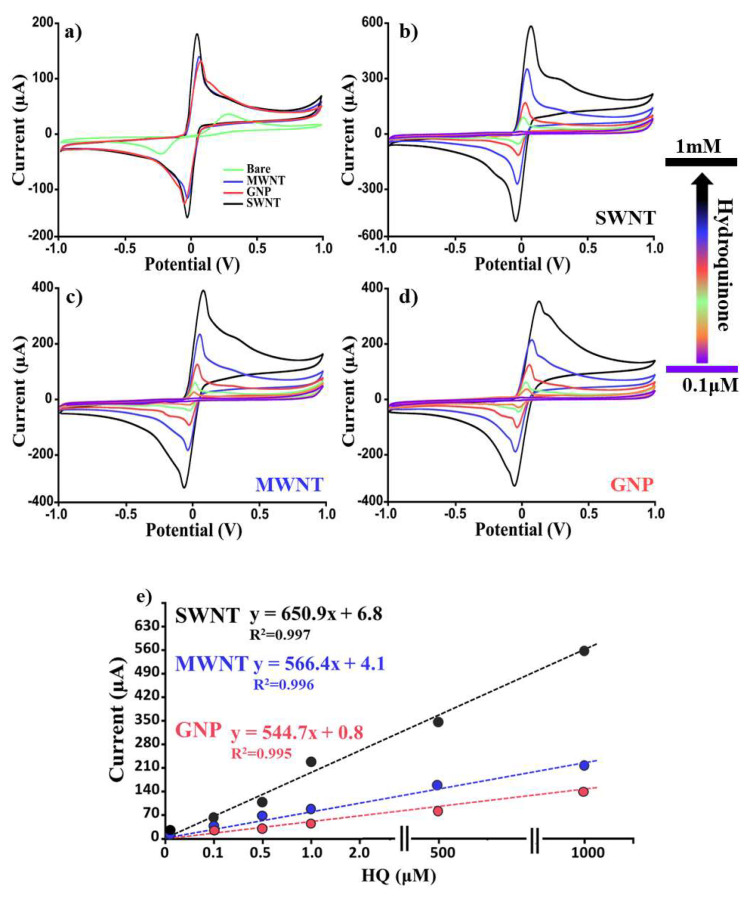
Dose–response voltammograms. Comparison of the CVs obtained by analyzing with all our platforms (SWNTs, MWNTs, GNPs, and bare SPEs) (**a**) the same concentration of HQ (5 mM) and (**b**–**d**) several HQ concentrations (from 0.1 µM to 1 mM); (**e**) relative linear regressions. Curves of one representative CNM-modified SPE sensor of at least 6 analyzed platforms are presented.

**Figure 6 nanomaterials-12-01779-f006:**
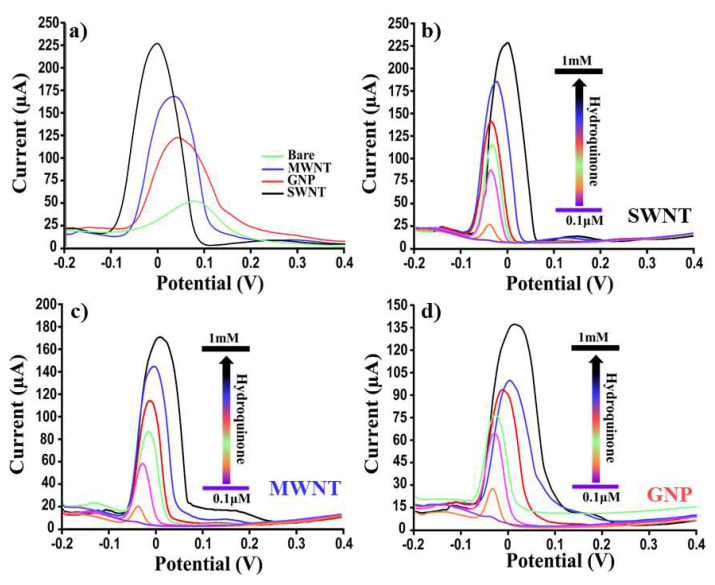
Dose–response voltammograms. Comparison of the SWVs for different platforms (SWNTs, MWNTs, GNPs, and bare SPEs) (**a**) at the same concentration of HQ (10 mM) and (**b**–**d**) at several HQ concentrations (from 0.1 to 1 mM). Curves of one representative CNM-modified SPE sensor of at least 6 analyzed platforms are presented.

**Table 1 nanomaterials-12-01779-t001:** Cathodic and anodic peak current intensity and their ratio, electron transfer rate constant (k^0^), peak-to-peak separation (ΔV), LOD, sensitivity, reproducibility (RSD%), active surface area (A), diffusion coefficient (D_0_), and percentage increase estimated for bare and CNM-modified SPEs have been reported. All analytical parameters are obtained from cyclic voltammogram traces.

[Fe(CN)_6_] ^4−/3−^ (10 mM)	Bare Electrode	SWNT	MWNT	GNP
|Iap| [µA]	17 ± 2	325 ± 5	274 ± 4	186 ± 17
|Icp| [µA]	22 ± 1	321 ± 4	269 ± 5	181 ± 15
|Ipa|/|Ipc|	0.80	1.01	1.02	1.02
|k^0^ | [cm/s]	1.9 × 10^−5^	5.6 × 10^−3^	2.1 × 10^−3^	1.4 × 10^−3^
A [cm^2^]	0.13	0.58	0.43	0.33
ΔE [mV]	197	85	110	124
D_0_ [cm^2^/s]	1.7 × 10^−7^	4.6 × 10^−6^	3.3 × 10^−6^	3.5 × 10^−6^
LOD [μM]	34.8	1.5	5.4	9.4
Sensitivity [mA/M cm^2^]	12	8.3	4.5	6.5
Reproducibility	10	5	3	5
% Increase |I_pa_|	/	258	294	276
% Increase |I_pc_|	/	480	580	312
|Iap| [µA]	17 ± 2	325 ± 5	274 ± 4	186 ± 17

**Table 2 nanomaterials-12-01779-t002:** Summary of the Limit of Detection (LOD) and reproducibility obtained using CV and SWV in the analyses of several concentrations of HQ and BQ in PBS solution. All analytical parameters are obtained from CV and SWV traces.

		Bare Electrode	SWNT	MWNT	GNP
**HydroQuinone (HQ)**					
CV	LOD (μM)	90.3	8.5	11.6	15.8
	Reproducibility (RSD%)	15	4	6	7
SWV	LOD (μM)	12.3	0.04	0.07	0.3
	Reproducibility (RSD%)	17	5	8	8
**BenzoQuinone (BQ)**					
CV	LOD (μM)	95.7	9.3	12.3	17.5
	Reproducibility (RSD%)	15	5	5	8
SWV	LOD (μM)	17.3	0.05	0.08	1.4
	Reproducibility (RSD%)	18	6	7	9

**Table 3 nanomaterials-12-01779-t003:** Summary of the Limit of Detection and reproducibility obtained using CV and SWV in the analyses of several HQ and BQ-spiked water solutions. All analytical parameters are obtained from CV and SWV traces.

		Bare Electrode	SWNT	MWNT	GNP
**HydroQuinone (HQ)**					
CV	LOD (μM)	334.5	80.3	13.7	166.6
	Reproducibility (RSD%)	17	8	9	9
SWV	LOD (μM)	132.3	2.2	1.7	5.3
	Reproducibility (RSD%)	17	10	10	10
**BenzoQuinone (BQ)**					
CV	LOD (μM)	275.5	1.4	102.5	108.5
	Reproducibility (RSD%)	14	10	10	10
SWV	LOD (μM)	121.2	1.7	2.9	4.3
	Reproducibility (RSD%)	18	11	12	11

**Table 4 nanomaterials-12-01779-t004:** Recovery study conducted on spiked-HQ water solutions.

	Spiked HQ Concentration C_S_ (μM)	Recovered HQ Concentration (C_T_–C_0_) (μM)	Recovery % (n = 6)	RSD% (n = 6)
SWNT	100	85	85.0	10
	200	177	88.5	9
	400	374	93.5	7
	800	743	92.8	6
MWNT	100	35	5.0	20
	200	129	64.5	17
	400	310	77.5	14
	800	633	79.1	12

## Data Availability

Not applicable.

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
