# Peer review of "Sensitive Detection of Industrial Pollutants Using Modified Electrochemical Platforms"

_nanomaterials, 2022, doi:10.3390/nano12101779_

Round 1

Reviewer 1 Report

Comments and Suggestions for Authors can be found  in the attachment.

Author Response

We would like to thank the reviewer for the constructive comments received. All the points raised by you have been addressed, as detailed in the following point to point reply list.

1.The structure of the manuscript needs to be adjusted. The chapter number needs to be corrected.

2-The expression needs to be strict, and the English grammar needs to be checked.

3.Please resize the image to look better.

Answer: Thanks for these comments. The manuscript has been improved in structure and quality of images, and language was carefully checked.

4. The background and significance of the study should be briefly described in the Abstract.

Answer: We have revised the abstract, highlighting the importance of the study and providing additional information to better underline the significance of the work.

5.Reduce the discussion of pollutant hazards in the Introduction. More discussion of current sensor developments should be provided

6.The innovation of this study needs to be further elaborated in the Introduction section.

7.In the Introduction, the discussion of experimental results should be reduced and the language should be simplified.

 Answer: Following these suggestions, we have revised and improved the introduction section. In particular, the introduction provides a more detailed discussion on the existing sensing techniques and the innovation over the current status introduced by this work.

8.Delete Line 202-204.

Answer: We deleted Lines 202-204 as requested.

9. Is the classical SPEs mentioned in Line 212 the same as the bare SPE in line 208. If so, please unify the way it is written.

Answer: Yes, they are the same platform, we unified as requested (bare SPE is now used everywhere)

10. As a sensor, response time is also an important parameter, is there any experiment about this parameter. If so, can the relevant parameters be provided?

Answer: Thanks for this important question: indeed, our platforms are designed to instantaneously determine the analyte of interest, therefore we agree on the importance of providing such a parameter. We have studied and optimized this parameter that can be estimated to be less than 2 minutes, thus providing a quick response.

11.Whether the amount of modification material is uniform and whether the amount of modification material used affects the electrochemical performances.

Answer: Thank you for the question.  The referee is right, since the amount of the modification  material influences the electrochemical performances of the electrode, as reported in literature (see Alex Yong Sheng EngChun Kiang ChuaM. Pumera (2015) Intrinsic electrochemical performance and precise control of surface porosity of graphene-modified electrodes using the drop-casting technique Electrochemistry Communications 59, 86-90). Therefore, in this work the concentration of the nanomaterials used to modify the electrode was optimised looking for the saturation amount able to ensure the reproducibility of the electrochemical performance of devices. A comment has been introduced in the paper to highlight this point (see rows 208-211)

12. The significance of the experimental results needs to be clarified in the Discussion section.

Answer: To address this suggestion, we have improved the introduction and discussion sections, with a better detailed analysis of the innovation proposed here and clear statements on the achieved results and performance.

13. Please check and unify the format of all the references in this manuscript.

Answer: Thanks for this comment, we have unified all the references.

Reviewer 2 Report

1) The conclusions can be shortened.

2) one of the major concerns is that more references related to the domain can be included.

R Ramaseshan, et al Journal of Applied Physics2007, 102 (11), 7,   The paper can be accepted for publication after answering those queries  

Author Response

We would like to thank the Reviewers for the constructive comments received.

All the points raised by you have been addressed, as detailed in the following.

1) The conclusions can be shortened.

 Answer: we ameliorated and shortened the conclusions as requested.

 2) one of the major concerns is that more references related to the domain can be included - R Ramaseshan, et al Journal of Applied Physics2007, 102 (11), 7,  

 Answer: we ameliorated and enriched the analysis of the literature, by adding 30 more references (including that suggested by the reviewer) and improving the introduction section

Reviewer 3 Report

The manuscript entitled "Sensitive Detection of Industrial Pollutants Using Modified Electrochemical Platforms" concerns the determination of hydroquinone and benzoquinone using SPE modified with carbon-based nanomaterials. In my opinion, the manuscript is interesting, well written and suitable for publication after small modifications.

  • In the introduction, it should be written anything about benzoquinone as is written for hydroquinone.
  • pag. 4 - lines 134 "Infrared analysis confirmed that..." Is it possible to add the FTIR spectra? for instance in the supplementary material.
  • pag. 6 Line 202-204 - should be removed
  • pag. 6 Line 242 - Figure 3B is not Figure 4B?

Is it not possible to include the voltammograms corresponding to the results in section 3.2.2 in the supplementary material?

Author Response

We would like to thank the Reviewers for the constructive comments received.

All the points raised by you have been addressed, as detailed in the following.

1)In the introduction, it should be written anything about benzoquinone is written for hydroquinone.

Answer: we ameliorated and improved the introduction, adding 30 more references.

2)Pag. 4 - lines 134 "Infrared analysis confirmed that..." Is it possible to add the FTIR spectra? for instance in the supplementary material.

Answer: we provided the FTIR spectra in the supplementary material, along with its description.

3)Pag. 6 Line 202-204 - should be removed

Answer: We removed it as requested.

 4)Pag. 6 Line 242 - Figure 3B is not Figure 4B?

Answer: We changed 3b to 4b.

5)Is it not possible to include the voltammograms corresponding to the results in section 3.2.2 in the supplementary material?

Answer: We provided the voltammograms in the supplementary materials as requested.

Round 2

Reviewer 1 Report

The revisions made by the authors are qualified and conscientious. So, I thought it could be published on Nanomaterials.